# Target Acquisition for Collimation System of Wireless Quantum Communication Networks in Low Visibility

**DOI:** 10.3390/e25101381

**Published:** 2023-09-25

**Authors:** Keyu Li, Tao Jiang, Yang Li, Xuemin Wang, Zhiqiang Zhan, Fengwei Chen, Zhengfu Han, Weidong Wu

**Affiliations:** 1CAS Key Laboratory of Quantum Information, University of Science and Technology of China, Hefei 230026, China; likeyu6969@163.com; 2Research Center of Laser Fusion, China Academy of Engineering Physics, Mianyang 621900, China; tjiang1397@126.com (T.J.); 18697293093@163.com (Y.L.); wangxuemin75@sohu.com (X.W.); zhiqiangzhan3@163.com (Z.Z.); chenfengwei2007@126.com (F.C.); 3School of Physical Sciences, University of Science and Technology of China, Mianyang 621900, China

**Keywords:** scattering, false alarm rate, detecting laser, trend line, intensity threshold, rising velocity threshold, target acquisition, information transmission and extraction

## Abstract

In severe low-visibility environments full of smoke, because of the performance degeneration of the near-infrared (NIR) collimation system of quantum drones communication networks, the improved dual-threshold method based on trend line analysis for long-wave infrared (LWIR) quantum cascade lasers (QCLs) is proposed, to achieve target acquisition. The simulation results show that smoke-scattering noise is a steeply varying medium–high-frequency modulation. At particle sizes less than 4 μm, the traditional dual-threshold method can effectively distinguish the target information from the smoke noise, which is the advantage of the LWIR laser compared to the NIR laser. For detecting lasers with high signal-to-noise ratios (SNRs), the method can achieve good target acquisition, by setting reasonable conventional thresholds, such as 0.7 times the peak intensity and 0.8 times the peak rising velocity. At low SNRs and steep intensity variation, the method can also achieve good target acquisition, by adaptively resetting new thresholds after filtering the detecting laser, such as 0.6 times the peak intensity and 0.6 times the peak rising velocity. The results of this paper will provide a reference for the performance improvement and refinement of the collimation system for wireless quantum communication networks in low visibility.

## 1. Introduction

Quantum key distribution (QKD), proposed by physicist Bennett and cryptographer Brassard in 1984 [1], is the BB84 protocol based on the principle of quantum mechanical measurements, and the E91 protocol [2,3] and B92 protocol [4] have been developed on its basis. QKD fundamentally guarantees the security of the key, which is very important. Based on QKD, the quantum communication network is a star-earth network based on “fiber optic node communica + star-earth communication”. Up till now, the network has made much progress in security, robustness and adaptability [5,6,7]. And it has realized many milestone leaps in communication distances, from 32 cm to 508 km [8], 509 km [9], 833.8 km [10] and 4600 km [11].

However, because of the inherent transmission loss of optical fiber and the constraints of the fixed orbits of satellites, the star-earth quantum communication network cannot meet all the needs of future quantum network construction. For example, for networks with a small range (e.g., communication between users in the same city), the star-earth network is less cost-effective, but a wireless quantum communication network of unmanned aerial systems (UAS) can compensate for these shortcomings, because drones are highly mobile and flexible [12]. Therefore, the wireless quantum communication network is a useful supplement to the two main quantum communication modes mentioned above.

In the field of wireless quantum communication networks, Nanjing University made significant progress in optical-relayed entanglement distribution using drones as mobile nodes in 2021 [13], and the team used multiple drones to form a multi-node relay mobile quantum drone network, which will hopefully fill the gap between satellite and ground-based quantum networks. Using such portable quantum nodes, the existing satellites can be connected to ground-based nodes, and this is expected to produce a long-range and extensive wireless quantum network, which will provide a novel technical way to break the technical bottleneck of “fiber wired optic transmission” for quantum communication. It is a milestone for realizing long-range wireless quantum communication. A review article in *Physics*, a scientific journal, is entitled “Quantum Drones Take Flight” [14]. The review states that, in order to achieve the transmission of entangled photons over longer distances, it is necessary to overcome the losses caused by the diffraction of light itself, which requires the construction of a laser collimation system. The review article points out the importance of the collimation system in quantum drone networks. Because the light source of the existing collimation system is a laser diode with a wavelength of 940 nm, its collimation performance is excellent in atmospheric environments with visibility up to tens of kilometers. But, once operating in low-visibility environments (e.g., visibility as severe as 10 m) filled with various aerosols (particle size 1 μm to 15 μm)—such as smoke, or smoke and soot—the 940 nm laser wavelength is less than, or even much less than, the aerosol particle size, so the transmission penetration distance, the information detection and the extraction of the NIR laser are severely limited, and the detecting laser is easily interfered with by the scattering of aerosols, resulting in false alarms or missed alarms [15]. The collimation performance will drop sharply, and the collimation system among multiple drones will not be able to achieve accurate target acquisition, pointing and tracking (APT), i.e., the drone network will become a scattered disk. In this way, the quantum drone network will also not be able to achieve point-to-point accurate alignment, to send and receive quantum communication information, and the advantages of quantum communication will eventually cease to exist. The Nanjing University research team also explained that the current experimental results were obtained under clear weather conditions, and expressed the hope that quantum information will be sent without the influence of weather conditions in the future [13]. Therefore, the APT system for collimating targets in quantum drone networks under low-visibility smoke aerosol interference needs to be further improved, in order to adapt to all-weather conditions in the future [13].

In recent decades, at low visibility, the anti-smoke aerosol interference for detecting the laser of a collimation system has been one of the difficulties in the field of laser detection, which is especially prominent in the field of laser fuzes. Because there is no relative reference in the field of wireless quantum communication networks, we can draw on relevant research in the field of laser fuzes, to address the improvement and refinement of the APT system in quantum drone networks. During laser transmission in the smoke and target reflection process, the non-uniformity of the smoke scattering and target reflection will distort the target information, accompanied by strong noise modulation. The problem—how to understand the transmission law of information and effectively extract the target information—belongs to the scope of generalized information theory. Of course, due to the randomness, uncertainty and fast-changing characteristics of smoke distribution, the transmission of detecting lasers can only be understood from the perspective of probability statistics. It is impossible to understand the transmission of detecting lasers in smoke, as in the laser directional transmission in geometric optics. For a detecting laser with strong scattering noise modulation, to achieve the acquisition and extraction of target information in a high-velocity and effective way, the novel detection system and the unique mathematical processing method have been research hotspots and the focus of laser fuzes in the field of information theory. Up to now, researchers have actively explored the detection regime and signal processing methods.

In the detection regime of target acquisition, in 2014, Wei Bin, Wang Wei and others designed a pseudo-random code laser fuze, using its own autocorrelation to achieve ranging and anti-interference performance [16]. In 2017, Li Zheng, Wang Feng and others proposed dual-wavelength laser fuzes—that is, the use of two wavelength radiation sources of violet and infrared light for target acquisition [17]. Duan Yabo proposed the detection technology of frequency-modulated continuous-wave (FMCW) laser and radio combination fuzes, studied the anti-interference theory of composite fuzes in a complex environment and the theory of height acquisition of targets in the detection area, gave full play to the advantages of the two detection regimes and improved composite fuze anti-interference capability [18]. Wei Zhang established a Monte Carlo simulation model of FMCW laser transmission in fog, analyzed comparatively the characteristics of the target echo signal and the fog echo signal of FMCW laser fuzes and proposed a fog backscattering suppression method based on the normalized frequency spectrum threshold of beat signals [19,20]. In 2018, Xie Shaoyu, Ye Mao et al. proposed polarized laser fuzes, i.e., a new technical method of laser fuze detection target acquisition, using the difference in the decoupling degree of the scattering of the polarized laser on the target and target background [21]. In 2020 and 2021, Lijuan Gao [22] and Li Lekun [23] proposed narrow pulse laser fuzes, proving that a narrow pulse laser fuze can effectively suppress smoke-detecting lasers, improve the fuze’s ability to identify targets in smokey environments, improve the fuze’s anti-smoke interference capability and reduce fuze false alarm.

In the signal processing methods of target acquisition, in 2007 Song Lei and Chen Shaohua proposed a multiple processing technique combining automatic gain control, pulse shape acquisition, and digital signal processing, which can better eliminate the interference of smoke smoke [24]. Zheng Yang [25] in 2014 and Lu Ming [26] in 2018 proposed a dual-threshold method based on the different characteristics of scattered signals generated by smoke- and target-detecting lasers. In 2013, Li Weiheng and Song Chengtian carried out research on simulation technology of the process with detection, scan and imaging, which was used in laser imaging fuzes for tank targets [27]. In 2018, Wang Xiaoju, Ma Heng et al. proposed a laser grayscale imaging fuze technology applicable to air-to-air missiles [28]. Based the grayscale profile image accumulated line by line by using the progressive target acquisition algorithm, the refined detection of laser fuzes could be achieved and their anti-interference capability could be improved [28]. In 2021, in the process of high-speed missile–target intersection, Qian Tang and Wei He used the deep migration learning method on the target recognition task of laser imaging fuzes [29], and Bing-ting Zha proposed a laser fuze specific target recognition algorithm tailored to small sample sizes, based on gray system theory [30]. In 2022, based on the difference of waveform features between targets and smoke, Meng Xiangsheng and Li Jing et al. proposed an anti-interference method of array laser waveform feature acquisition [31]. Based on the combination of the improved Harris plus the smallest univalue segment assimilating nucleus (SUSAN) corner detection algorithm and rectangularity, in 2023, Zhou Yu and He Wei were able to efficiently and accurately filter out suspended particles, such as smoke, to reduce the impact on the operation of laser imaging fuzes [15].

In recent years, based on multilayer InGaAs/InAlAs or GaAs/AlGaAs materials grown by molecular beam epitaxy (MBE) technology, mid-infrared QCLs have been maturing. The miniaturization and portability of QCLs make them very suitable as detection sources for various ship-based, airborne and vehicle-mounted detectors (see Figure 1), and they are increasingly used in various fields [32,33,34]. Our team has also developed a miniaturized QCL (see Figure 1). Based on the dual advantages of the LWIR QCL wavelength (8–14 μm) being much larger than the NIR wavelength and its strong penetration ability in smoke, and on the trend line analysis method often used in the field of stocks, and combining the dual advantages of the detection regime and signal processing methods of target acquisition, the improved dual-threshold method based on trend line analysis for LWIR QCLs is proposed for target collimation of wireless quantum communication networks—that is, using the LWIR QCL instead of the NIR laser as a new circular-viewing detection light source, using the mercury cadmium tellurium (MCT) detector developed by our team to detect the LWIR QCL signal, and then using the improved dual-threshold method based on a trend line of “rising firstly and then falling” to identify and extract target information. This offers a new idea and a new technology for the anti-interference of smoke aerosol in low-visibility environments and for drone target acquisition. Target acquisition is the foundation, the key and the core of the APT system, and it is also the focus and difficulty of the APT system, in terms of achieving the collimation function. Target pointing and tracking after the target acquisition is a matter of water. Therefore, this paper focuses on the target acquisition function of the APT system of wireless quantum communication networks.

## 2. The Transmission Characteristics of Detecting Laser in Smoke Aerosols

In a low-visibility environment, the particle sizes of aerosols are close to or even exceed the wavelength of the detecting laser, resulting in a strong scattering noise signal in the detecting laser. The noise easily makes the collimation system generate false alarms or missed alarms, thus issuing wrong instructions and causing the collimation system to fail.

### 2.1. Model of Scattering in Smoke

According to the Mie scattering theory, when the aerosol particle size is close to or even larger than the wavelength of light, its scattering is
(1)$$P_{j}\left(R\right)=P_{0}S_{r}f\left(\theta \right)\sigma_{S}\int_{D_{0}}^{R}\frac{S_{R}\left(r\right)}{r^{2}}e^{-2\sigma_{S}\left(r-D_{c}\right)-k\left(R-r\right)}dr,$$
where $P_{j}\left(R\right)$ is the scattering noise power, *R* is the optical range of the detecting laser, $P_{0}$ is the initial power of the detecting laser, $S_{r}$ is the pupil area of the detector, f(θ) is the scattering phase function of the aerosol, $\sigma_{S}$ is the Mie scattering coefficient of the aerosol, $D_{C}$ is the distance between the detector and the smoke, $D_{0}$ is the blind depth of the collimation system, wave number $k=2\pi /\left(c\tau \right)$, *c* is the light velocity and τ is the pulse width of the detecting laser.

Drawing on the LIDAR distance equation [35], the target detecting laser intensity $P_{T}$ is
(2)$$P_{T}=\frac{P_{0}T^{2}\rho D^{2}\eta_{t}\eta_{r}}{4\pi \beta^{2}R_{CT}^{2}},$$
where *T* is the atmospheric transmission coefficient, which is calculated according to different visibility, ρ is the target reflectivity, *D* is the effective incident aperture of the detector, $\eta_{t}$ and $\eta_{r}$ are the optical transmission efficiencies of the laser transmitter and the laser receiver, β is the laser divergence angle and $R_{CT}$ is the distance between the collimated system and the target.

The total detecting laser $P_{r}$ of the detector—namely, the detection echo information power—is the sum of the scattering noise $P_{j}$ and the target information $P_{T}$. When detecting the presence or absence of a target based on the detecting laser, four different scenarios occur, as follows:(1)The probability that a target appears and is judged correctly to be present is the “probability of acquisition”, and it is expressed as $P_{d}$;(2)The probability of the appearance of the target with the incorrect judgment that there is no target is the “probability of omission”, and it is expressed as $P_{la}$;(3)The probability of the target not appearing and of judging correctly that there is no target is the “probability of correct non-acquisition”, and it is expressed as $P_{an}$;(4)The probability that the target does not appear and is judged to be incorrect is the “false alarm probability”, and it is expressed as $P_{fa}$. The false alarm probability is calculated as follows:
(3)$$P_{fa}=\frac{N_{TP}-N}{N_{TP}},$$
where $N_{TP}$ denotes the number of detected targets and *N* denotes the number of actual targets.

### 2.2. Interference Characteristics of Aerosols to Detecting Laser Information

Aerosol scattering is closely related to the aerosol particle size. In order to simplify the analysis, in calculation it is necessary to assume a uniform distribution of smoke and a fixed value of particle size, and to assume that the aerosol and the target are fixed, with only the collimation system in motion. Of course, this assumption does not exist in the actual environment: the purpose of such an assumption is to better study the law of aerosol interference with the detection of the laser and the transmission characteristics of detecting the laser in the smoke aerosol.

Based on the miniaturized QCL and MCT detector developed by our team (see Figure 1), the specific parameters used in the simulation calculation of Figure 2 are as follows: the QCL emission wavelength is 8.9 μm; the pulse radiation average power *P* is 10 mW; the initial power $P_{0}$ of the detecting laser is PPΓPDFΓPDF with the pulse duty factor $\Gamma_{\mathrm{PDF}}$; the beam waist width of the light source ω=3 mm; the laser dispersion angle β is 5 mrad; the distance $R_{CT}$ between the collimation system and the target is 30 m; the chip aperture of the MCT detector is 0.25×0.25 mm^2^; the effective incident aperture *D* is 10 mm; the target aperture is tens of meters (i.e., the detecting laser has a full target reflected signal in one pulse cycle); and the reflectivity ρ is 10%; the optical transmission efficiency $\eta_{t}$ and $\eta_{r}$ of the detector transmitter and receiver are 0.9; the aerosol particle size gradually increases from 1 μm to 10 μm; and the atmospheric visibility V=10 m. The atmospheric transmission coefficient *T* is mainly calculated according to the atmospheric visibility and the following empirical formula:(4)$$T\;=\;1-\alpha_{\mathrm{atten}}\left(V\right)\times R_{CT}/1000,$$
where, in a low-visibility environment (e.g., visibility as severe as 10 m), the attenuation coefficient of $\alpha_{\mathrm{atten}}\left(V\right)$ is 2 to 4 per one kilometer.

Figure 2 shows that the power of the detecting laser received by the detector gradually decreases as the aerosol particle size increases. When the particle size is less than or equal to 3 μm (Figure 2a), the difference between the target information intensity and the noise intensity is very obvious, the SNR of the detecting laser is at a high level, and it is obvious that the intensities of the second and fourth pulses with the presence of a target are much higher than the noise signal. Therefore, the target is found easily at the moment of these two pulses, and the target information and smoke can be well identified by the traditional intensity threshold method at this time. However, when the particle size reaches 3.87 μm (Figure 2b), the smoke reflection noise in the detecting laser has risen sharply to half of the target information intensity, which can very easily cause a false alarm. For example, if the threshold value is selected as 60% (i.e., 0.064 μW) of the maximum value of 0.106 μW, the first and third pulses caused by smoke scattering have also reached the threshold, which will lead to the false alarm of the APT system and to the collimation performance failure. When the particle size reaches 3.97 μm (Figure 2c), the intensity of the detecting laser is already smaller than that in Figure 2a, but the intensity of the smoke-scattering noise is already comparable to that of the target information, and all five pulses have reached the threshold, so it is basically impossible to distinguish between noise and target information. At this point, by simply using the traditional intensity threshold method, the correct extraction of target information can no longer be achieved, and the target acquisition method must be further optimized.

Therefore, the transmission law of detecting QCL information in smoke aerosol can be understood in this way: due to the scattering difference of different particle sizes on the detecting laser, there will be different intensity modulation of smoke noise in the detecting laser, and the modulation intensity will approach or even exceed the target information intensity. Because the collimation system, the target and the smoke aerosols are all in motion, basically, and the particle sizes of non-uniform distribution smoke are random values, the smoke noise is a steeply varying medium–high-frequency modulation. Therefore, in Section 4, the smoke noise will be discussed according to the two following cases: a high SNR and steep intensity variation without medium–high-frequency modulation nose (shown in detail in Section 4.2); and a low SNR with steep intensity variation and medium–high-frequency modulation nose (shown in detail in Section 4.3).

## 3. Improved Dual-Threshold Method Based on Trend Line Analysis for an LWIR QCL Detecting Laser

The physical process of the improved dual-threshold method is that the intensity threshold and the trend line analysis of “rising firstly and then falling” in a latter pulse period are used firstly: with the satisfaction of the two above conditions, the rising velocity threshold method is then used. When the pulse trend and the two thresholds—namely $P{}_{i}\left(t\right)⩾P{}_{0}$ and $P{}_{v}\left(t\right)⩾P{}_{t}$—have come true, it is confirmed to have found the target. The workflow of the improved dual-threshold method is shown in Figure 3.

As shown in Figure 3, the first step is the setting of the cycle period parameter *T* and the judgment of the sampled time point *t*. If t⩽T, the two thresholds will be used. Otherwise the collimation system will wait for the command of the next detection.

The second step is target detection, by using the improved dual-threshold method. For every sampled time point, *t*, there are three following cases.

Case No.1 is the filter judgment for the detecting laser. If there is a low SNR, a steep intensity change and medium–high-frequency noise modulation in the detecting laser, it is necessary to filter first, after which, the detection laser intensity $P_{i}\left(t\right)$, the rising velocity $P_{v}\left(t\right)$ of intensity $P_{i}\left(t\right)$ and the new thresholds of P0 and Pt (shown in detail in Section 4.3) are calculated. On the other hand, there is no need to filter, and the four physical quantities—namely, $P_{i}\left(t\right)$, $P_{v}\left(t\right)$ and the conventional thresholds of P0 and Pt (shown in detail in Section 4.1 and Section 4.2)—are calculated directly, which is conducive to saving time for high-velocity detection.

Case No.2 is the judgments of the intensity threshold and the trend line. When the signal intensity is equal or greater than the intensity threshold—namely, $P{}_{i}\left(t\right)⩾P{}_{0}$—the trend line analysis of the “rising firstly and then falling” method is used, to identify the target signal or noise signal authenticity. If the intensity threshold time point is the target signal, in a latter pulse period the pulse trend must be “rising firstly and then falling”, and the intensity threshold time point pulse width is taken as the target acquisition time point of the collimation system. On the other hand, the intensity threshold time point must be a steep change point and noise signal because of dissatisfaction with the pulse trend of “rising firstly and then falling” in a latter pulse period. Therefore, the steep intensity change point that reaches the intensity threshold but does not satisfy the specific pulse trend is set to zero (i.e., actively deleted), and the sampled time point *t* is changed into the next point t+1.

Case No.3 is the judgment of the rising velocity threshold, by using the rising velocity threshold method to discriminate whether the rising velocity of the signal reaches the rising velocity threshold point—namely, $P{}_{v}\left(t\right)⩾P{}_{t}$. According to Lu Ming’s study, the rising velocities of smoke noise over the ocean and the target information are 3.0 dB/ms and 22.2 dB/ms, respectively [26], and there is an essential difference between the two rising velocities. If the rising velocity threshold point does not appear, the next sampled time point—namely, t+1—is calculated.

The third step is target acquisition. When the three conditions are satisfied—namely, the pulse trend of “rising firstly and then falling” in a latter pulse period, the two threshold points $P{}_{i}\left(t\right)⩾P{}_{0}$ and $P{}_{v}\left(t\right)⩾P{}_{t}$—the collimation system will obtain the accurate information of “find the target” (shown in detail in Section 4).

## 4. Simulation and Verification

### 4.1. Traditional Dual-Threshold Method Analysis

Based on the flow of Figure 3, the rising velocity of the detecting laser was calculated for Figure 2b,c, and the results are shown in Figure 4. Figure 2a is already discriminated by the traditional intensity threshold method, so no further analysis is required.

In Figure 4, it is shown that when the aerosol particle sizes are 3.87 μm and 3.97 μm, respectively, there is not much difference in the rising velocity of the detecting laser, because the intensities of the noise and the target information are similar, i.e., the smoke aerosol scattering signal and the target signal are no longer distinguishable when the aerosol particle size is getting larger. However, based on the rising velocity difference of the target information and smoke noise—namely, using the offline experiment in advance, to determine and set the reasonable rising velocity threshold in the collimation system—then, for example, with the reasonable rising velocity of $7.99\times 10^{-3}$
μW/ns in Figure 4a, only the second and fourth pulses in Figure 4a can be determined as the target information, but the five pulses in Figure 4b are all smoke-aerosol-scattering noise. Therefore, there are some limitations to the rising velocity threshold method.

Combining Figure 2 and Figure 4, the variations of false alarm rate with threshold and aerosol particle size are calculated under different discriminating conditions, and the results are shown in Figure 5 and Figure 6, respectively.

In Figure 5, it can be seen that the reasonable setting of the conventional threshold value is very important for the target acquisition. If the setting is too large, it is easy to generate missed reports and miss the target; if the setting is too small, the determination results of both the intensity threshold method and the rising velocity threshold method are incorrect, due to the influence of noise, and it is easy to generate false alarms. When the conventional threshold value is set reasonably, such as 0.7 times the intensity peak and 0.8 times the rising velocity peak (see Figure 5), the traditional dual thresholds can identify the target and can minimize the false alarm rate (as shown in detail in Section 4.2).

In Figure 6, it can be concluded that, with the increase of aerosol particle size, the noise impact from aerosol particle scattering gradually increases. When the particle size is larger than 3.8 μm, the false alarm rate increases sharply if the intensity threshold method or the rising velocity threshold method is used alone, i.e., it is difficult to identify the noise and the target, while the false alarm rate can be reduced to zero when the particle size is smaller than 4 μm, by using the traditional dual-threshold method, which can effectively distinguish the smoke noise and the target information (see Figure 4 and Figure 6). When the particle size is smaller than 4.2 μm, the false alarm rate can also be reduced, to a certain extent; however, when the particle size is larger than 4.2 μm, the traditional dual-threshold method is also limited (see Figure 6). Therefore, the traditional dual-threshold method must be further optimized.

### 4.2. High SNR and Steep Intensity Variation without Medium–High-Frequency Modulation Noise, Using Improved Dual-Threshold Method

According to the theory of section No.2, when the particle size is close to 3.87 μm (see Figure 2b) the noise will likely approach the target information intensity gradually. As shown in Figure 7, as the aerosol and the target are moving in the actual environment, it is more likely that the scattering noise of the large-particle-size smoke aerosol in the detecting laser is a kind of transient steep change at a certain moment. In the following discussion, we firstly consider the target acquisition under a high SNR steep change of the detecting laser: for example, an SNR of 30 (see Figure 7a), where the two conventional thresholds are 0.7 times the intensity peak and 0.8 times the rising velocity peak, based on Figure 5 and Figure 6.

Figure 7a shows that of the six steep intensity changes of the detecting laser, five are very obvious—the exception being the fourth 349 ns moment, which is a slight noise attached to the target information—and the six steep changes have exceeded the intensity threshold, so the intensity threshold method is no longer applicable. In Figure 7b, it is shown that if the rising velocity threshold method is used, the rising velocities of the detecting laser at the six steep changes are larger than those of the target information, and the rising velocity threshold method is also no longer applicable. The red dashed line in Figure 7c shows that if the traditional dual-threshold method is used to find the target acquisition moment in the collimated system, it will result in an error conclusion. All six steep changes, except the fourth one, will be identified as the target acquisition moment, but the normal target information will be masked, which will lead to false alarms and wrong commands from the collimation system. When the detecting laser reaches the intensity threshold at a certain moment, the intensity analysis of “rising firstly and then falling” of the improved dual-threshold method can be used for signal authenticity acquisition—that is, within a pulse envelope after the threshold point, the trend of the signal intensity must be “rising firstly and then falling”, otherwise it will be judged as the intensity steep change point and noise signal, and the steep change point will be set to zero for the active deletion. After the intensity threshold method of discrimination, the rising velocity threshold method can be used for target acquisition. The black solid line in Figure 7c shows that the improved dual-threshold method can screen out these six steep change points, because the intensity steep changes, although satisfying the intensity threshold and the rising velocity threshold, do not satisfy the trend line characteristic of “rising firstly and then falling” within one pulse envelope. Therefore, the improved dual-threshold method can accurately find three target acquisition moments (see the three black solid lines in Figure 7c).

### 4.3. Low SNR with Steep Intensity Variation and Medium–High-Frequency Modulation Noise Using Improved Dual-Threshold Method

According to the theory of section No.2, when the particle size is larger than 3.97 μm, the detecting laser will contain some intensity steep points and a low SNR noise (see Figure 8).

As shown in Figure 8a, the smoke noise intensity will likely gradually become greater than the target information intensity, such as the noise intensity of the sixth 558 ns moment. Therefore, the SNR of the detecting laser will be relatively low: for example, an SNR of 3.

In this case, the rising velocity of the six steep variations is greater than the signal rising velocity at the target (see Figure 8b), and the traditional dual-threshold method will find three of the six steep variations as the moment of target acquisition for the collimation system. If the improved dual-threshold method is used, only two moments of target acquisition moment can be found and the first pulse beside the fourth noise is masked by the noise modulation, i.e., there are three target pulses but only two can be found (see the two black solid lines in Figure 8c). And this omission will cause the collimation system to find the target one pulse cycle later, which will greatly reduce the “high-velocity collimation” performance of the collimation system in quantum communication networks.

For detecting lasers with a low SNR and steep intensity variation, both traditional and improved dual-threshold methods fail, because the target information is masked, due to excessive noise; therefore, filtering before target acquisition can be considered, and it is necessary. Using a fast filtering method, such as improved gradient particle swarm optimization (IGPSO, described in another paper), the detecting laser information in Figure 8 is filtered, and the processed results are shown in Figure 9.

As shown in Figure 9a, after filtering, the noise and intensity steep change points have been wiped out, basically, so the traditional and improved dual-threshold method can achieve a good target acquisition effect (see Figure 9c). There are three target acquisition time points by using the improved dual-threshold method, but there are six target acquisition time points by using traditional dual-threshold method, because of the limitations of it. This shows that the improved dual-threshold method has an advantage over the traditional dual-threshold, because its results get close to the practical situation. Moreover, there is a difference of a pulse-width delay of target acquisition moment point in the two methods, which is because the target acquisition time point is the intensity threshold time point plus the width of a single pulse in the improved dual-threshold method. However, the filtered image is definitely distorted, and the intensity threshold and the rising velocity threshold must be redetermined by the adaptive method, to find all the target acquisition moments. After finding the three maxima of the three pulse envelopes in Figure 9a, the intensity threshold is set to 0.7 times the average of the three maxima, which ensures that all target information intensities are above the intensity threshold, and the rising velocity threshold is set in the same way, i.e., the two new thresholds are 0.6 times the peak intensity and 0.6 times the peak rising velocity, which is different from the “0.7 times the peak intensity and 0.8 times the peak rising velocity” at the high SNR detecting laser. Of course, from practical considerations of the collimation system “high-velocity response” effect, the detecting laser filtering is time-consuming, and filtering is not considered as a last resort, because the filtering will increase, or even delay the discrimination time of the collimation system, which is very unfavorable to the practicality of the collimation system.

## 5. Conclusions

Aiming at the problem that the collimation performance of wireless quantum communication networks will drop sharply in smoke environments with low visibility, the new method and technique of a new collimation system is proposed. These are the LWIR QCL and MCT in the new system. Based on the trend line analysis of “rising firstly and then falling” in a latter pulse period for detecting laser, the improved dual-threshold method is proposed for target acquisition. After analysis, it was shown that smoke noise will add steeply varying medium–high-frequency modulation to the target acquisition information. When the particle size is less than 4 μm, the LWIR detecting laser can effectively be distinguished between smoke noise and target information by using a traditional dual-threshold method. In the case of a high SNR and steep intensity modulation, by setting a reasonable conventional threshold, such as 0.7 times the peak intensity and 0.8 times the peak rising velocity, the improved dual-threshold method can achieve good target acquisition, but the traditional dual-threshold method will probably lead to a great false alarm. For a low SNR and steep intensity modulation of the detecting laser, the improved dual-threshold method can also achieve the target acquisition, by adaptively resetting new thresholds after filtering the detecting laser, such as 0.6 times the peak intensity and 0.6 times the peak rising velocity.

The above conclusions show that the LWIR laser is better than the NIR laser, in terms of the mechanism of anti-smoke interference for target acquisition of quantum drones communication networks. Therefore, the development of the LWIR QCL collimation system is a wise choice in conditions of smoke interference. The new collimation system will provide a way forward for collimation performance improvement and refinement of quantum drone communication networks in smoke environments with low visibility, provide reference for thoroughly solving the performance degradation of the collimation system, provide a theoretical basis for realizing the target acquisition function of APT systems in low visibility, and lay a certain foundation for the promotion and application of LWIR QCL. The hope is held out by the Nanjing University research team [13] that quantum information will be sent without the influence of weather conditions in the future, and that the concept of the collimation system put forward by Michael Schirber [14] will become a reality. 

## Figures and Tables

**Figure 1 entropy-25-01381-f001:**
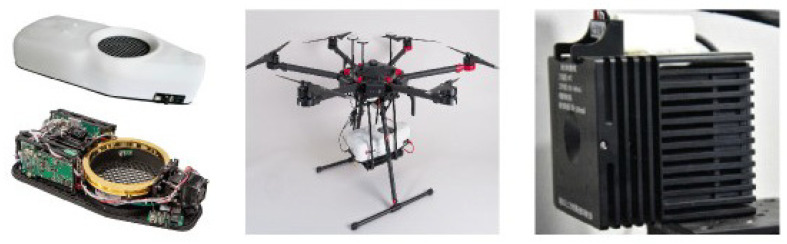
Airborne portable ${\mathrm{CH}}_{4}$ laser spectrometer mainframe using a QCL (**left**), a spectrometer on-board system (**middle**) [32] and the wavelength of 8.9 μm and 3D dimensions of a 70×70×70 mm^3^ micro QCL developed by our team (**right**).

**Figure 2 entropy-25-01381-f002:**
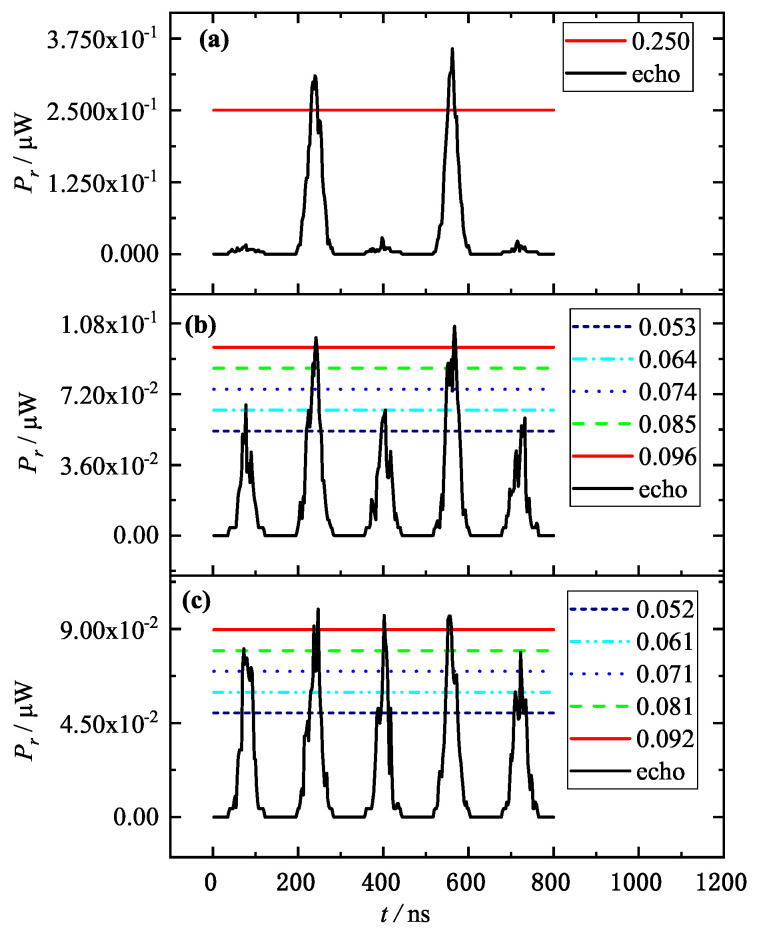
Detection echo information power $P_{r}$ for five pulse periods at random appearances of the target and horizontal lines for different intensity thresholds with aerosol particle sizes of 3 μm (**a**), 3.87 μm (**b**) and 3.97 μm (**c**).

**Figure 3 entropy-25-01381-f003:**
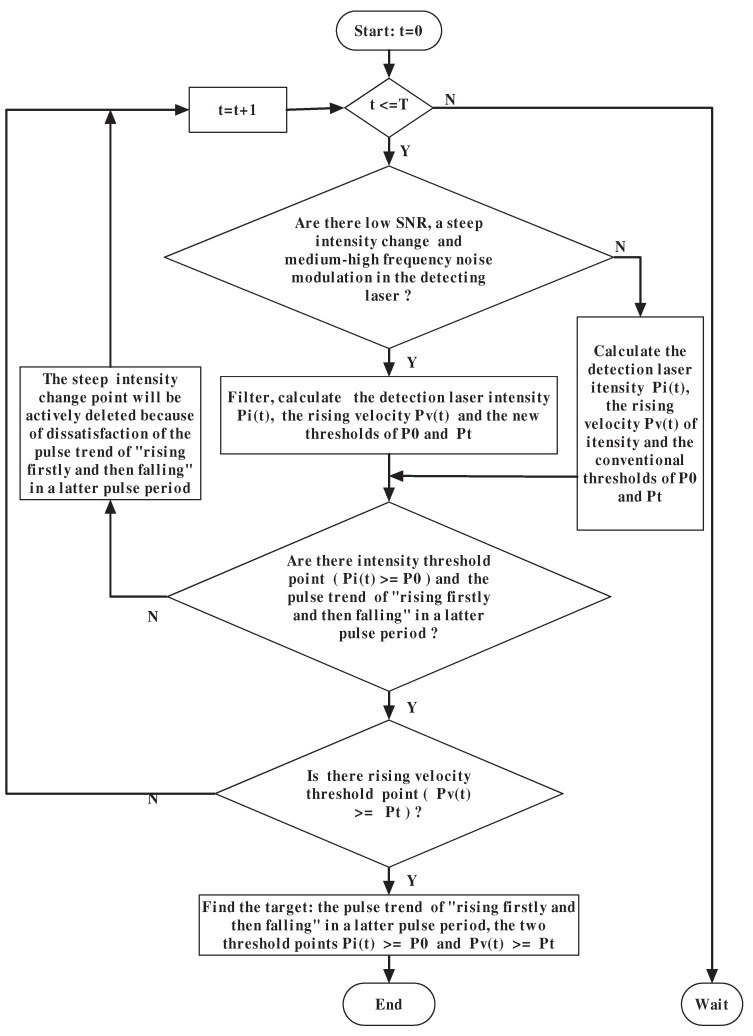
Flow chart of the improved dual-threshold method on trend line analysis for an LWIR QCL detecting laser.

**Figure 4 entropy-25-01381-f004:**
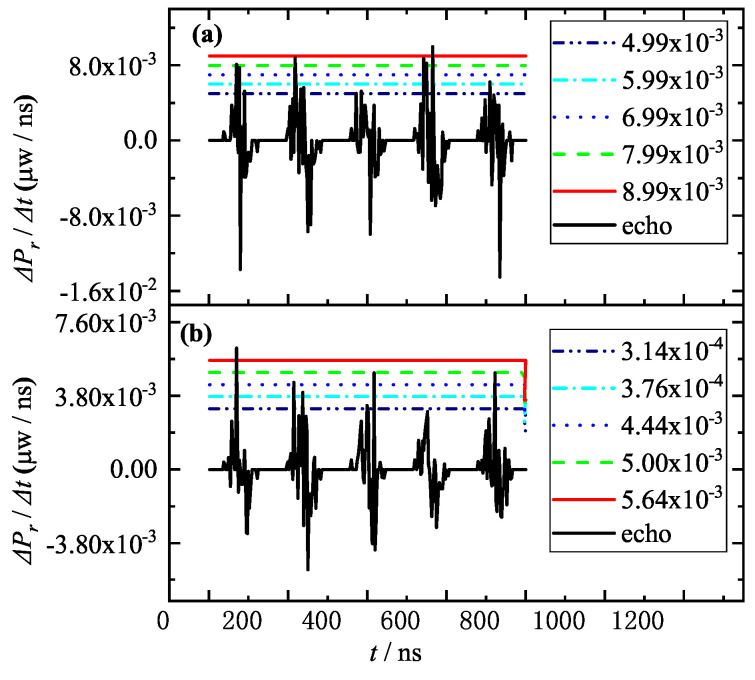
Variations of the rising velocity $\Delta P_{r}/\Delta t$ of the detecting laser echo and the horizontal lines of different rising velocity thresholds for the random appearance of the target in five pulse cycles with aerosol particle sizes of 3.87 μm (**a**) and 3.97 μm (**b**).

**Figure 5 entropy-25-01381-f005:**
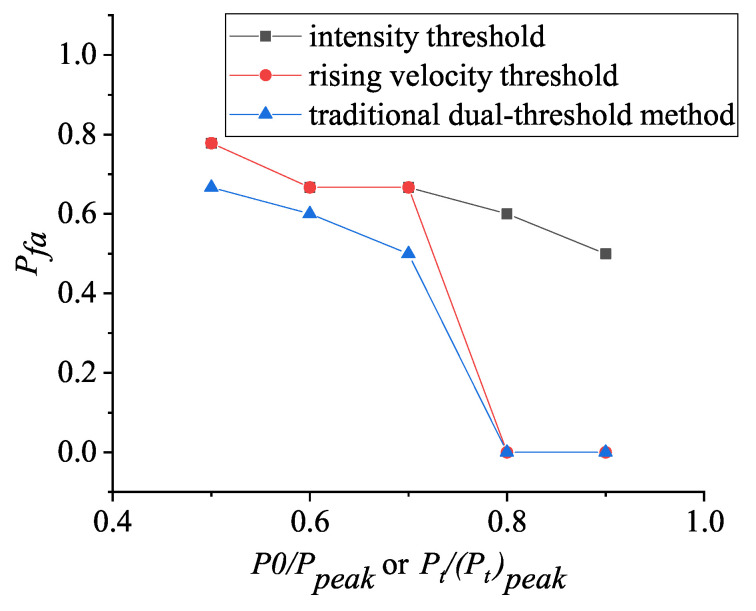
Variations of false alarm rate $P_{fa}$ with threshold ratio for aerosol particle size of 3.97 μm, based on three discrimination methods.

**Figure 6 entropy-25-01381-f006:**
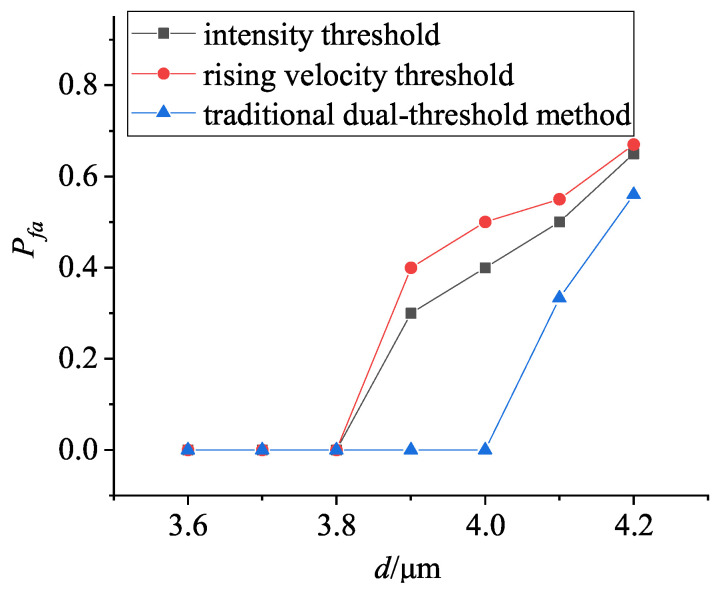
Variations of false alarm rate $P_{fa}$ with particle size based on three methods with traditional dual-threshold method of 0.7 times the peak intensity and 0.8 times the peak rising velocity.

**Figure 7 entropy-25-01381-f007:**
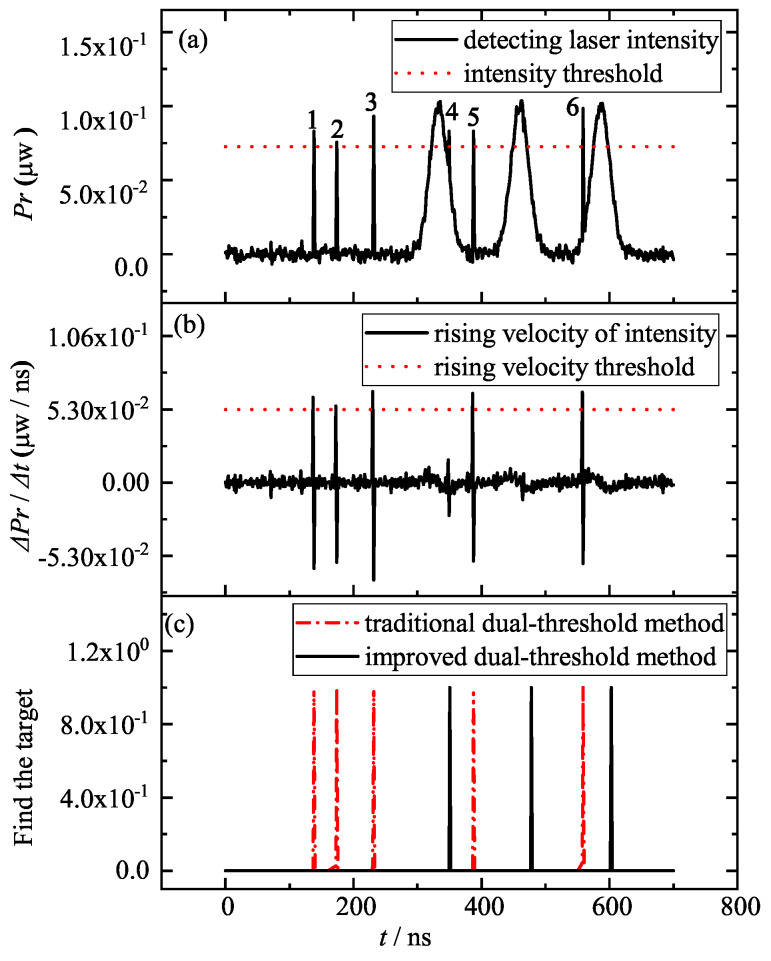
Detecting laser intensity $P_{r}$ (**a**), rising velocity $\Delta P_{r}/\Delta t$ (**b**) and the moment of target acquisition for the collimated system with traditional and improved dual-threshold methods ((**c**), “target acquisition” is denoted by 1) for a high SNR and six steep variations of the detecting laser, where t is 138 ns, 174 ns, 232 ns, 349 ns, 387 ns and 558 ns, with six detecting laser intensity steep changes.

**Figure 8 entropy-25-01381-f008:**
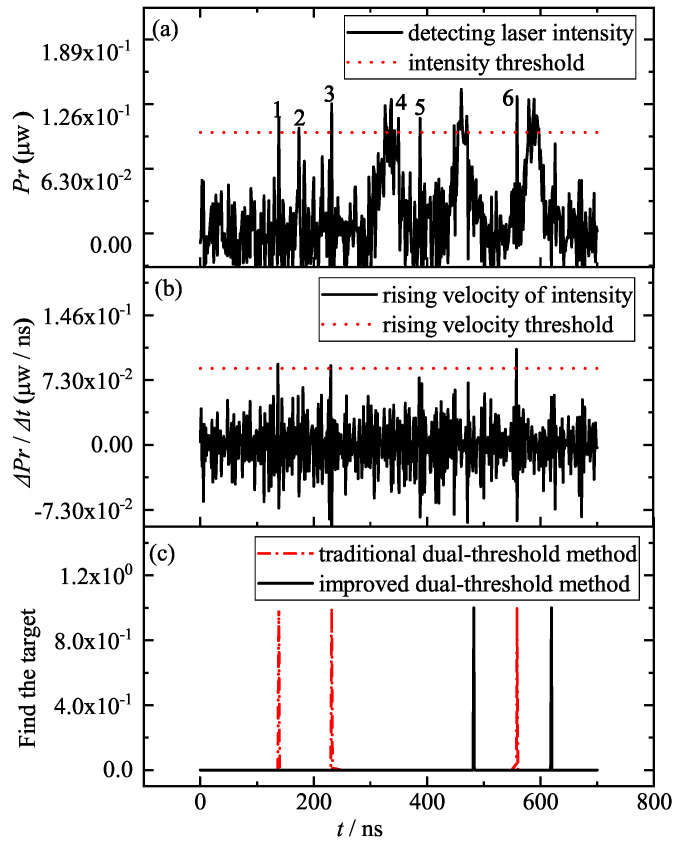
Detecting laser intensity $P_{r}$ (**a**), rising velocity $\Delta P_{r}/\Delta t$ (**b**) and the moment of target acquisition for the collimated system with traditional and improved dual-threshold method (**c**) for a low SNR and six steep variations of the detecting laser.

**Figure 9 entropy-25-01381-f009:**
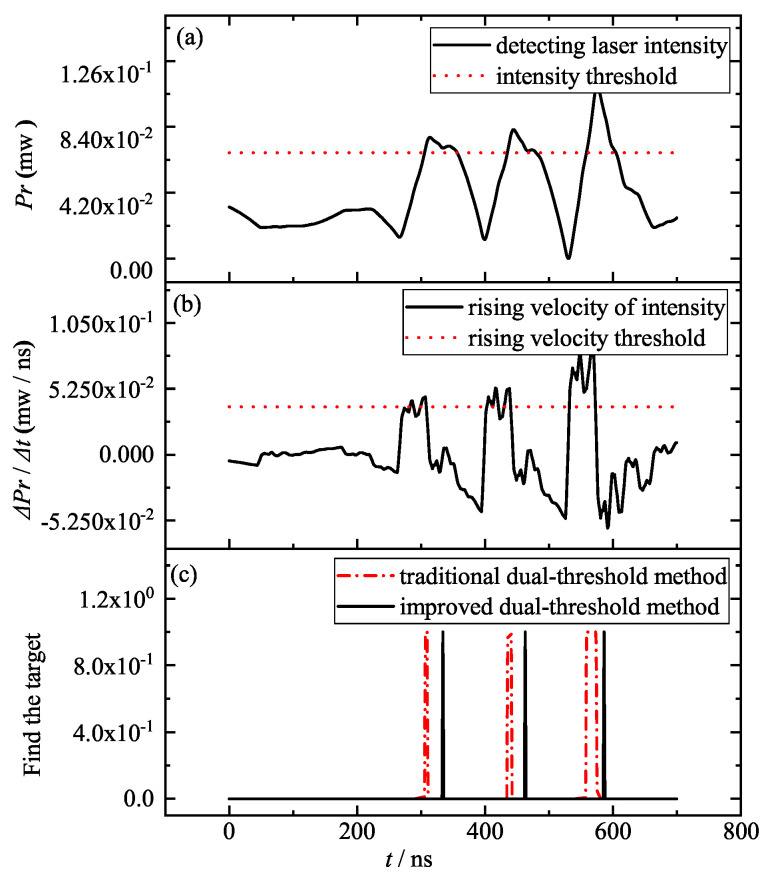
Filtered detecting laser intensity $P_{r}$ (**a**) for Figure 8a, rising velocity $\Delta P_{r}/\Delta t$ (**b**) and the moment of target acquisition for the collimated system with traditional and improved dual-threshold method (**c**).

## Data Availability

Not applicable.

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
