# Peer review of "Target Acquisition for Collimation System of Wireless Quantum Communication Networks in Low Visibility"

_entropy, 2023, doi:10.3390/e25101381_

Round 1

Reviewer 1 Report

In this paper, the authors proposed the improved dual-threshold method based on trend line analysis for long-wave infrared (LWIR) quantum cascade laser (QCL) is proposed to achieve target acquisition. The results of this paper will provide a reference for the performance improvement and refinement of collimation system for wireless quantum communication networks in low visibility. I think the scheme and the results are interesting and would perhaps worth publication after some necessary revisions. The technical points I would like to be considered in the revised version are the following:

1.
As far as I know, atmospheric transmission coefficient is related to factors such as temperature and altitude. How these factors are defined in this paper should be explained in detail. Are the parameters set in section 2,2 generic?

2.
There is little literature listed in this paper. The authors should make this seriously if they want to publish their work in a serious scientific journal.

3.
The figure 7, 8, and 9 in the text needs to be carefully revised.

Minor editing of English language required

Author Response

Thank you for your letter and the referee’s comment regarding our manuscript, which is entitled "Target Acquisition for Collimation System of Wireless Quantum Communication Network in Low Visibility".

We are grateful to the reviewers for their recognition of our manuscript. He (She) thinks that some necessary revisions can be improved in our manuscript. We have thought about this carefully and made changes according to the referee’s suggestions. We believe these issues have been addressed in the revised version.

We greatly appreciate the referee’s reading and comments. Those comments are all valuable and helpful for revising and improving our paper, as well as the important guiding significance to our research. We have addressed his (her) in the revised version, and we believe the manuscript now offers additional clarification and insight to readers.

Here, we include a new version of the manuscript, a list of changes, and a detailed response to the referee. The revised parts of our manuscript have been highlighted in red.

Reviewer 2 Report

To tackle the performance failure of near-infrared (NIR) collimation system of quantum drones communication network, an improved dual-threshold method based on trend line analysis for long-wave infrared quantum cascade laser (QCL) is proposed to achieve target acquisition.

Some meaningful conclusions are reached.

It is original and interesting, and I would like to recommend its aceptance for publication in ENTROPY after mandatory revision.

The motivation and contributions can be clarified.

Moderate editing of English language would be better. Some sentences are too long to easily understand.

The results should be explained more clear, especislly from the aspect of theory.

The conclusion should be briefly rearranged.

Moderate editing of English language would be better. Some sentences are too long to easily understand.

Author Response

Dear Editor and referees,

Thank you for your letter and the referee’s comment regarding our manuscript, which is entitled "Target Acquisition for Collimation System of Wireless Quantum Communication Network in Low Visibility".

We are grateful to the reviewers for their recognition of our manuscript. He (She) thinks that some necessary revisions can be improved in our manuscript. We have thought about this carefully and made changes according to the referee’s suggestions. We believe these issues have been addressed in the revised version.

We greatly appreciate the referee’s reading and comments. Those comments are all valuable and helpful for revising and improving our paper, as well as the important guiding significance to our research. We have addressed his (her) in the revised version, and we believe the manuscript now offers additional clarification and insight to readers.

Here, we include a new version of the manuscript, a list of changes, and a detailed response to the referee. The revised parts of our manuscript have been highlighted in red.

Sincerely,

Ke-yu Li

(On behalf of all authors)
